# Low-Cost Sensor Network for Air Quality Assessment in Cabo Verde Islands

**DOI:** 10.3390/s24237656

**Published:** 2024-11-29

**Authors:** Anedito Zico da Costa, José P. S. Aniceto, Myriam Lopes

**Affiliations:** 1CICECO-Aveiro Institute of Materials, Department of Chemistry, University of Aveiro, 3810-193 Aveiro, Portugal; azcosta@ua.pt (A.Z.d.C.); joseaniceto@ua.pt (J.P.S.A.); 2Centre for Environmental and Marine Studies (CESAM), Department of Environment and Planning, University of Aveiro, 3810-193 Aveiro, Portugal

**Keywords:** air quality, low-cost sensors, air pollution, Cabo Verde, PM2.5, PM10, NO_2_

## Abstract

This study explores the application of low-cost sensor networks for air quality monitoring in Cabo Verde islands, utilizing Clarity Node-S sensors to measure fine particulate matter with diameters equal to or smaller than 10 µm (PM10) and 2.5 µm (PM2.5) and nitrogen dioxide (NO_2_) gasses, across various locations. The sensors were strategically placed and calibrated to ensure coverage of the whole archipelago and accurate data collection. The results consistently revealed seasonal patterns of dust variation across the archipelago, with concentrations of particulate matter exceeding World Health Organization (WHO) limits in all regions. However, Praia frequently exhibits the highest levels of air pollution, exceeding a 200 µg/m^3^ daily average, particularly during the dry season. Seasonal variations indicated that pollutants are significantly higher from November to March due to Saharan dust flux (a phenomenon locally know as *Bruma Seca*). Other cities showed more stable and lower pollutant concentrations. This study highlights the potential of low-cost sensors to provide extensive and real-time air quality data, enabling better environmental assessment and policy formulation. However, the variability in equipment accuracy and the limited geographical coverage remain the main limitations to be overcome. Future research should focus on these issues, and a sensor network integrated with reference methods could be a great asset to enhance data accuracy and improve outcomes of air quality monitoring in the country.

## 1. Introduction

Atmospheric pollutants, capable of long-range transport and formation, present multiple challenges in their management and mitigation, affecting extensive areas [1]. The quality of air assumes significant importance in the islands of Cabo Verde, located in the Atlantic, off the African coast [2]. The peculiar geography of the archipelago, characterized by strong winds and sparse vegetation, makes it particularly susceptible to high levels of air pollution, with direct impacts on public health and sustainable development. Studies conducted on the Sahara dust that reaches Cabo Verde have shown that this phenomenon, locally known as *Bruma Seca*, significantly affects air visibility and population comfort [3]. The archipelago constantly faces the presence of dust flows, with a noticeable increase in events between October and March. Specifically, December and January record the most intense flows of mineral dust originating from North Africa [4]. This seasonal pattern of dry haze implies significant challenges for air quality management and the implementation of environmental and public health protection measures in Cabo Verde. The exposure to high levels of particulate matter (PM2.5 and PM10) is associated with a variety of health problems, including respiratory and cardiovascular diseases [5]. The frequent episodes of high air pollution, as evidenced by the graphs, can lead to an increase in hospitalizations and mortality associated with these conditions. Nitrogen dioxide (NO_2_) is a known respiratory irritant, and elevated levels are correlated with an increase in chronic respiratory diseases and acute respiratory tract infections [6].

Previous studies in Cabo Verde, such as those conducted at the Cabo Verde Atmospheric Observatory (CVAO) on São Vicente Island, have focused on the properties and origins of aerosol particles and cloud condensation nuclei (CCN) near sea level and at cloud level. These studies have concluded that Saharan dust and sea salt dominate the PM10 particle composition in near-surface air masses [7,8,9,10]. While the Cabo Verde archipelago has recently established an air quality monitoring network, there has been limited research utilizing low-cost sensor (LCS) networks for comprehensive air quality monitoring in the region.

Traditionally, air quality monitoring has depended on reference stations which, despite providing high-precision data, are associated with high operational and maintenance costs, representing a significant limitation for their expansion [11], especially in developing countries [12]. In this context, LCS emerges as a promising solution for obtaining a more extensive set of air quality data in communities at a much lower expense compared to existing stationary, high-precision reference stations [13]. The development of these sensors has revolutionized Air Quality Monitoring (AQM), since they have lower energy requirements and are less expensive compared to reference equipment, while showing an increasing quality. These sensors can then be installed in multiple locations where reference monitoring is not possible, providing wider spatial coverage—one of the greatest advantages of LCS [14]. Although promising due to their accessibility and flexibility, they face criticism for variability in the accuracy and reliability of the data collected, critical elements for informed decision-making [15].

Clarity Node-S sensors are among the most advanced low-cost sensors available today. They are designed to measure particulate matter (PM2.5 and PM10) and NO_2_, both of which are critical indicators of air quality due to their significant health impacts. High levels of PM2.5 and PM10 can lead to respiratory and cardiovascular diseases, while NO_2_ exposure is associated with adverse effects on the respiratory system [16,17]. These sensors [18] have been successfully deployed in various projects worldwide, demonstrating their adaptability and effectiveness in diverse environmental conditions. Kortoçi et al. [19] used low-cost sensors for monitoring air quality in urban environments and validated the LCS data against high-end reference monitoring stations. Zareba et al. [20] used low-cost sensor data to successfully predict air pollution using machine learning techniques. Their capacity to provide real-time, localized air quality data has made them an essential tool for both urban and rural monitoring, helping policymakers and public health officials address air pollution challenges more efficiently. By offering a cost-effective solution, these sensors have expanded air quality monitoring networks, allowing for more comprehensive data collection and analysis across different regions.

This study aims to explore the applicability and efficacy of low-cost sensor networks in air quality monitoring in Cabo Verde. It intends to assess their feasibility as environmental assessment tools while also contributing to the scientific field in Cabo Verde. This work not only seeks to enrich existing knowledge on local atmospheric pollution but also hopes to offer strategic insights for the formulation of more effective air quality management policies. The implementation of low-cost sensor networks may enable continuous and real-time monitoring of air quality, covering various locales within the archipelago, thus promoting a more informed and dynamic environmental response.

## 2. Materials and Methods

### 2.1. Study Area Description

Cabo Verde is an archipelagic country, comprising ten islands—Santo Antão, São Vicente, Santa Luzia, São Nicolau, Sal, Boa Vista, Maio, Santiago, Fogo, and Brava—and five main islets (Branco, Raso, Luís Carneiro, Grande, and de Cima), with a total surface area of approximately 4.033 km^2^ (Figure 1). It is located between 14°23′ and 17°12′ north latitude and 22°40′ and 25°22′ west longitude from Greenwich [21]. According to the National Adaptation Plan of Cabo Verde [22], the country features an arid to semi-arid climate, is hot and dry, with an average annual temperature of around 25 °C. During the cooler months from January to April, the temperature averages about 20 °C and during the warmer periods of the year, from August to October, temperatures exceed 27 °C. The annual relative humidity fluctuates between 60 and 85%, with average precipitation values for the arid coastal zones being less than 100 mm, as seen in the islands of Sal, Boa Vista, and Maio. For the mountainous islands, the average precipitation can reach values around 600 mm, as is the case of Santiago, Fogo, and Santo Antão islands. However, recent observations have shown a significant reduction on average, especially in the last four years. The 2021 census shows that Cabo Verde has a resident population of 491.233 inhabitants, with approximately 74% living in urban areas and 25% in rural areas [21].

### 2.2. Sensor Network Description

The infrastructure of this study consists of a network of Clarity Node-S sensors, selected for their ability to monitor a broad range of atmospheric pollutants, specifically PM10, PM2.5, and NO_2_. The sensor deployment was strategically planned to ensure the coverage of all main urban areas (in the main inhabited islands), with sensors installed on the busiest streets of each city and with ease of access to facilitate hardware maintenance, but avoiding local hot spots related to specific sources like fairs and constructions sites. The devices were positioned at approximately 4 meters above ground level and around 1 meter from the road. All sensors were calibrated according to international standards by the suppliers, ensuring the reliability and accuracy of the data collected. The exact locations of the sensors are provided in Table 1.

This device is engineered for the accurate measurement of atmospheric contaminants, with a particular focus on the parameters of particulate matter (PM10, PM2.5) and NO_2_. It employs Laser Light Scattering technology for the detection of particulate matter, featuring a detection range from 0 to 1000 µg/m^3^ and a resolution of 1 µg/m^3^. For the measurement of NO_2_, it utilizes an Electrochemical Cell with remote calibration, spanning a range of 0 to 3000 parts per billion (ppb), with a resolution of 1 ppb (Figure 2). Table 2 shows the main characteristics of sensors implemented in the network.

Beyond these parameters, the Node-S can assess additional critical metrics, including the number concentration of PM1, PM2.5, and PM10 particles. This system is also equipped with internal sensors for the measurement of temperature and relative humidity.

### 2.3. Data Collection Methods

To ensure a comprehensive and representative assessment of air quality in the Cabo Verde Islands, the data collection methodology was planned to cover the entire year from January to December 2023 and the entire archipelago, while being easily accessible for hardware maintenance, thus guaranteeing minimal missing data. Our monitoring period enabled us to analyze seasonal variations and the impact of different weather conditions on pollutant levels. A network of strategically placed Clarity Node-S sensors enabled us to capture regional differences in air composition, providing the high temporal resolution crucial for identifying pollution peaks and analyzing daily and hourly patterns. Each sensor was programmed to record readings at 15-min intervals, ensuring continuity and consistency in data collection throughout the study period. 

Additionally, meteorological data were independently obtained to support and contextualize the air quality measurements. These meteorological datasets were sourced from the National Institute of Meteorology and Geophysics (INMG), which operates dedicated weather stations located at key strategic points, including several airports throughout the 3 islands (Sal, São Vicente, and Santiago). The meteorological data included variables such as temperature, wind speed, and wind direction, all of which are critical factors that can influence air pollutant levels and their dispersion patterns. By integrating air quality data with meteorological conditions, we aim to provide a more complete understanding of the factors that affect air quality in Cabo Verde.

The data collected by the sensors are transmitted in real time to a centralized server through an internet connection, ensuring that the latest readings are always available for immediate processing. This automated data transmission system not only enhances the efficiency of data collection but also minimizes the data loss or corruption that can occur with manual data handling.

For the processing and analysis of these data, a suite of digital data analysis and processing tools were employed, specifically Excel, Python [23]—with the libraries pandas, datetime, NumPy, and matplotlib—and R [24], with the libraries Openair, Openairmaps, and ggplot2, chosen for their powerful data manipulation capabilities and extensive support libraries.

## 3. Results and Discussion

### 3.1. Meteorology

Figure 3 shows monthly temperature profiles for Praia, Mindelo, and Santa Maria over a year, with Praia exhibiting the greatest fluctuations, ranging from 22 °C to 34 °C, and a peak in November. Mindelo and Santa Maria have more stable profiles, with temperatures ranging from 22 °C to 30 °C and 22 °C to 28 °C, respectively, and peaks during the middle of the year. Measurements from the meteorological stations at the airports in Praia, Sal, and São Vicente show similar temperature profiles, with most of the peaks registered on the same day, reflecting comparable climatic conditions. These typical seasonal variations in the region show temperature increases during the hottest months (July to September) and significant drops at the end of the year. The most pronounced fluctuations and highest temperature peaks in Praia can be attributed to specific local factors, such as urban density, urbanization and intense human activities, geographical characteristics, or eventual drift in the equipment.

The rainfall pattern in Cabo Verde in 2023, illustrated in Figure 4, shows a prolonged dry season from January to June, followed by a brief and intense rainy season between August and September. A peak is reached in September, with the islands of Fogo and Santo Antão recording monthly accumulations of over 400 mm, according to the graphic. This pattern is consistent with the earlier observations of Correia [25], who identified a marked seasonal division between the rainy season and the prolonged dry period. Correia analyzed data from 1941–1990 and reported that, in the islands of Santo Antão, Santiago, and Fogo, rainfall is concentrated in the months of August to October, with September being the wettest. These patterns have also been confirmed by recent data from the Tropical Rainfall Measuring Mission project, which uses remote sensing to monitor rainfall in tropical areas with limited pluviometry networks, such as Cabo Verde [26]. This seasonal behavior tends to intensify the negative effects of pollution during the dry months; studies such as those by McMullen et al. [27], Ouyang et al. [28], and Wang et al. [29] show that precipitation events tend to decrease the concentration of particles in the atmosphere.

The winds over Cabo Verde (Figure 5) have a dominant northeasterly pattern, strongly influenced by the North Atlantic trade winds. The islands of Mindelo, Praia, and Sal, located at different points in the archipelago, are characterized by winds of varying intensity, but with a consistent direction. In Mindelo, the winds are more intense, often exceeding 50 m/s, while in Praia and Sal the average intensity varies between 10 and 30 m/s. The predominance of north-easterly winds facilitates the transportation of dust from the Sahara, which can aggravate the levels of suspended particles and affect public health [9].

### 3.2. Pollution Profile

Analysis of the pollution roses in Figure 6, for Mindelo, Praia, and Santa Maria, reveals that the PM2.5, PM10, and NO_2_ are predominantly transported by wind coming from the northeast (NE). This pattern is consistent with the trade winds that bring dust from the Sahara to the archipelago [3]. In Mindelo, although the direction of entry of pollutants is also from the northeast, the intensity of the winds is stronger, which should facilitate the dispersion of particles and keep pollutant concentrations at relatively low levels. In Praia, where the winds are less intense, there is a greater accumulation of pollutants, with significantly high concentrations of PM2.5 and PM10, which reach maximum levels. Santa Maria presents an intermediate scenario, with moderate concentrations due to reasonable ventilation, but less intense than Mindelo. These patterns indicate that in Mindelo, where wind speeds are higher (see Figure 5), the dispersion of pollutants is more effective, leading to improved air quality compared to Praia, where the accumulation of particles is more pronounced.

It was also observed that larger cities, such as Praia and Mindelo, tend to record higher concentrations of PM2.5 and NO_2_. This can be attributed to a combination of natural and anthropogenic factors, such as intense traffic and industrial and commercial activities. For example, Praia, with a population of over 142,000 inhabitants, shows the highest annual average concentrations of PM2.5 (24.16 µg/m^3^) and NO_2_ (18.12 µg/m^3^), significantly above the levels observed in smaller population cities like Porto Novo and Sal Rei. Table 3 provides an annual average concentration for all studied locations in Cabo Verde, focusing on three key pollutants: PM2.5, PM10, and NO_2_.

The spatial analysis of PM2.5 concentrations in Cabo Verde, shown in Figure 7, suggests a possible correlation between population and air pollution levels. In the city of Praia, PM2.5 concentrations reach values close to 24 µg/m^3^, which may be associated with more intense urban activities, such as road traffic. In contrast, less populated areas, such as Mindelo, Santa Maria, and São Filipe, exhibit lower PM2.5 concentrations, ranging from 8 to 18 µg/m^3^. Studies conducted in other regions also indicate a positive relationship between population density and atmospheric pollutant concentrations. Borck and Schrauth [30] analyzed panel data from Germany and found that increased population density is associated with higher concentrations of NO_2_ and particulate matter. Kim et al. [31] studied the impact of three major socio-economic factors (income per capita, population density, and city population size) on PM2.5 concentrations in 254 cities across six developed countries, finding that a 1% increase in population density results in a 0.058% increase in PM2.5 concentrations. In this way, the high levels of PM2.5 observed in Praia can be partially explained by the concentration of urban activities as well as the previously mentioned lower dispersion capacity.

Figure 8 shows that there are small variations in pollutant concentrations between cities, except for Praia, which has the highest medians and a significant number of outliers. This may be explained by the higher population in Praia [32] compared to the other cities (see Table 3). Outliers, defined as values greater than 1.5 times the interquartile range, were identified in all locations, corresponding to concentration peaks. These outliers may represent episodic pollution events, such as construction work, or natural phenomena, such as *Bruma Seca*, as well as possible measurement errors in the sensors. There were dust events in Cabo Verde that coincided with significant increases in suspended particle concentrations, especially towards the end of the year, potentially exacerbating pollutant levels, particularly in the cities most exposed to the influence of dust-laden air masses from the Sahara Desert.

### 3.3. Time Series and Exceedances of the WHO Air Quality Guidelines (AQG) Levels

The evaluation of the air pollutant concentration data obtained from the sensor network revealed distinct temporal patterns. Figure 9 shows the daily averages of PM2.5, PM10, and NO_2_ concentrations for the sites studied during 2023, along with the World Health Organization (WHO) Air Quality Guidelines (AQG) level. The city of Praia consistently has the highest levels of pollutants, often exceeding the AQG levels recommended by the WHO. In contrast, the other cities, such as Mindelo, Porto Novo, Sal Rei, Santa Maria, and São Filipe, have more stable and significantly lower concentrations, generally within the AQG levels. Towards the end of the year, there is a notable spike in PM2.5 and PM10 levels. This may be attributed to seasonal meteorological conditions, such as dust storm in the Sahara region [33], which causes the occurrence of *Bruma Seca* [34], or increased human activities during the festive season, which may include higher traffic volumes and the increased use of heating sources. In the case of the *Bruma Seca* phenomenon, air masses carry high levels of particulate matter, causing occasional spikes in pollutant concentrations that exceed the recommended limits set by the WHO for air quality [16]. The elevated NO_2_ concentration in Praia may be associated with factors such as intense urbanization, as suggested by Borck and Schrauth [30], which affect wind circulation and the dispersion of pollutants, and the presence of motor vehicles, which are significant sources of nitrogen oxides. Additionally, the local geographical and meteorological configuration can influence the dispersion and concentration of NO_2_, increasing the levels observed in the city.

When observing PM10 concentrations (Figure 9b), a much smaller difference is noted between Praia and the other cities compared to PM2.5 and NO_2_ (Figure 9a and 9c, respectively). This can be considered normal due to several factors, such as the more diverse origin of PM10 particles, including Sahara dust, construction activities, and sources like sea spray. This broader range of sources can result in a more uniform distribution across different urban and semi-urban environments, as reflected in the strong correlations between PM10 concentrations across cities in the heatmap (Figure A1). Additionally, a significant peak in pollutant levels is observed at the end of the year, especially in PM2.5 and PM10 levels. This peak could be attributed to seasonal meteorological conditions, such as dust storms in the Sahara region [33], or increased human activities during the festive season, which may include higher traffic volumes and the increased use of heating sources.

Table 4 complements this analysis by summarizing the number of daily exceedances of AQG levels for PM2.5, PM10, and NO_2_ in all the locations studied. The city of Praia stands out with the highest number of exceedances for all pollutants, recorded on more than 330 days above the guideline for PM2.5 and NO_2_ during the year under review.

### 3.4. Seasonal Analysis

Figure 10 shows the monthly averages of air quality in 2023, highlighting two distinct seasons: the rainy season (August, September, October) and the dry season (remaining months) [22]. Praia consistently shows the highest concentrations of PM2.5, PM10, and NO_2_ throughout the year, especially during the dry season, suggesting higher local pollution due to traffic and industrial activities. During this time, PM10 frequently exceeds a 200 µg/m^3^ daily average, in contrast with the WHO AQG of 25 µg/m^3^. The other cities have more uniform and significantly lower concentrations. At the end of the year, there is a significant peak in PM2.5 and PM10 levels, probably due to increased human activity during the festive season and possible dust storms in the Sahara region.

Monthly averages during the dry season reveal worrying levels, with PM2.5 and NO_2_ concentrations around 40 µg/m^3^ and PM10 concentrations surpassing 60 µg/m^3^, particularly in December. These levels pose a potential health risk, namely in cardiorespiratory conditions, especially given the cumulative impact of prolonged exposure. Seasonal data show that there is a 15% reduction in PM2.5 and 16% reduction in PM10 during the wet season in Praia. However, the NO_2_ levels show only a slight decrease of 3.0%, showing that emission sources like vehicular emissions remain relatively consistent throughout the year. This analysis is further supported by Figure A2, which illustrates the seasonal differences in pollutant concentrations in the studied cities. The higher pollutant concentration values in the dry season reinforce the importance of the contribution of natural sources (*Bruma Seca*) to the observed air quality. The differences are not very great because even in the wet season the number of days and amount of rainfall in Cape Verde is quite low.

### 3.5. Daily Profiles and Weekly Profiles

Figure 11 shows the daily and weekly profiles of PM2.5, PM10, and NO_2_ concentrations in the cities studied. As shown before, the city of Praia presents the highest concentration levels for all pollutants, with significant peaks coinciding with commuting times in the morning, lunch time, and evening, due to intense road traffic and anthropogenic activities during these periods. At weekends, NO_2_ concentrations decrease by 8.35%, reinforcing the link between weekday human activities and pollution levels. In contrast, PM2.5 concentrations in Praia show a slight increase of 3.55%, which can be explained by the playful and cultural activities characteristic of this population. For example, it is very common for families and friends to get together at the weekend for parties and barbecues during weekends compared to weekdays.

Morning and evening peaks are also observed in the other monitoring sites, highlighting the importance of traffic emissions. Lunch time peaks observed also in São Filipe and Porto Novo can have contributions from the domestic combustion of biomass that is commonly used in Cape Verde for cooking [35].

## 4. Conclusions

This study assessed the application of a low-cost sensor network for air quality monitoring in Cabo Verde to measure PM10, PM2.5, and NO_2_ across various locations across the archipelago for a one-year period. Praia consistently had the highest levels of PM2.5, PM10, and NO_2_, often exceeding WHO recommendations, especially during the dry season. Praia exceeded the WHO Air Quality Guidelines limits regarding PM2.5 concentrations on 336 days throughout the year, while PM10 and NO_2_ concentrations were exceeded on 94 and 331 days, respectively. In contrast, the cities of Mindelo, Porto Novo, Sal Rei, Santa Maria, and São Filipe exceeded the WHO limits on few occasions, suggesting that urban and industrial activities in Praia, combined with a higher population, are significant contributors to the pollution levels observed.

Seasonal analysis highlights a variability in the pollutant concentration in Praia, with the dry season showing substantial increases in PM2.5 and PM10 concentrations. During this period, PM10 frequently exceeds a 200 µg/m^3^ daily average (WHO guideline: 45 µg/m^3^). Monthly averages during the dry season reveal concerning concentrations, with PM2.5 and NO_2_ around 40 µg/m^3^ and PM10 surpassing 60 µg/m^3^, particularly in December. These levels pose a potential risk to respiratory health, especially given the cumulative impact of prolonged exposure. The daily and weekly profiles of pollutant concentrations reveal distinct patterns, particularly in Praia, where peaks coincide with intense urban traffic, such as commuting hours, and reductions are seen on weekends. These trends are much less pronounced or even unseen in other cities.

The data open an opportunity to develop more in-depth studies, to anticipate peak episodes and study meteorological and anthropogenic influences on air quality more precisely. This approach could provide valuable input for formulating more effective environmental management and public health policies in Cabo Verde and other regions with similar characteristics. Low-cost sensor networks are a promising tool for monitoring air quality in this context. Although they have accuracy limitations compared to reference methods, they offer a practical and relatively inexpensive solution for collecting data in real time, allowing for a more widespread coverage compared to reference methods.

## Figures and Tables

**Figure 1 sensors-24-07656-f001:**
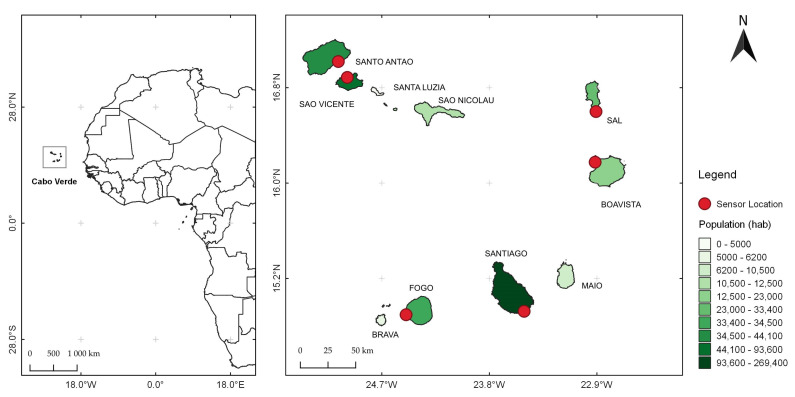
A map of the location of the Cabo Verde in relation to the African continent (left map) and a detailed map of the islands (right). The shades of green represent the population across the islands and red dots indicate the locations of air quality sensors.

**Figure 2 sensors-24-07656-f002:**
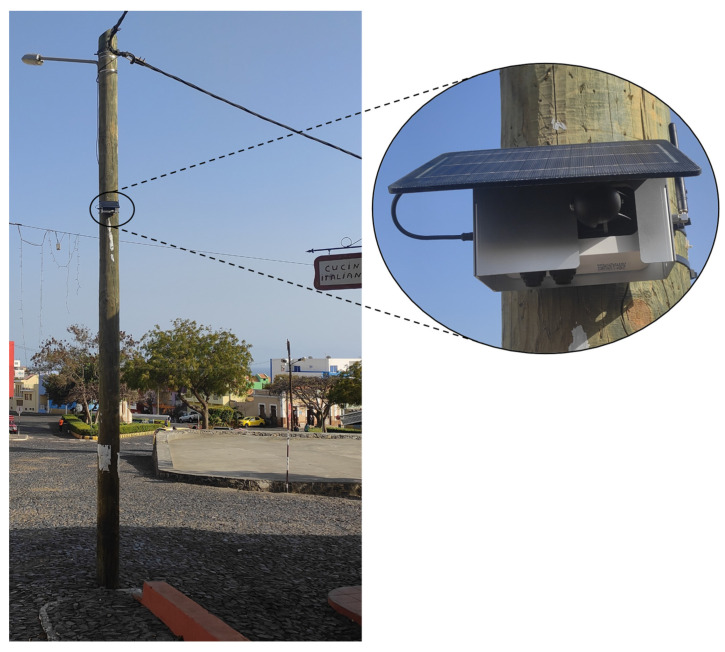
Clarity Node-S sensor installed in São Filipe city, on Fogo Island.

**Figure 3 sensors-24-07656-f003:**
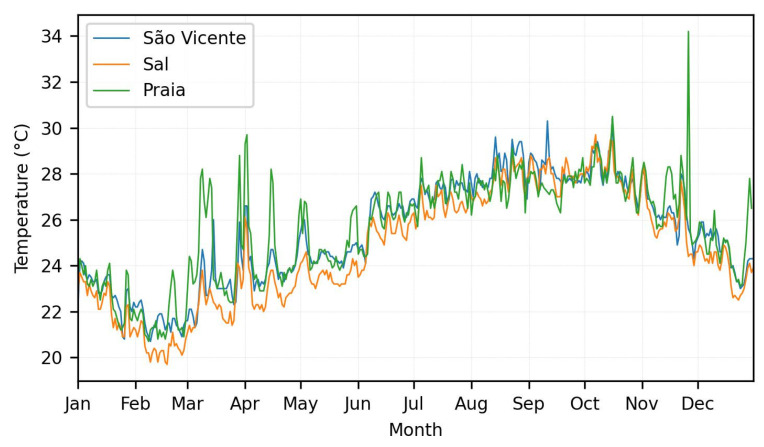
Daily average temperature in 2023 for Mindelo, Santa Maria, and Praia.

**Figure 4 sensors-24-07656-f004:**
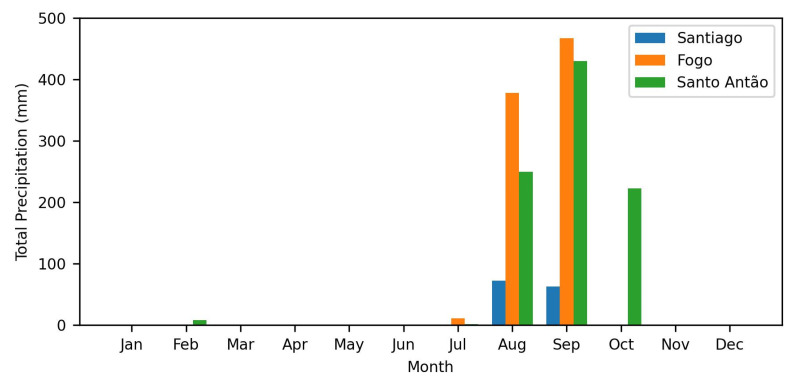
Total monthly precipitation (in mm) in Santiago, Fogo, and Santo Antão islands throughout the year of 2023.

**Figure 5 sensors-24-07656-f005:**
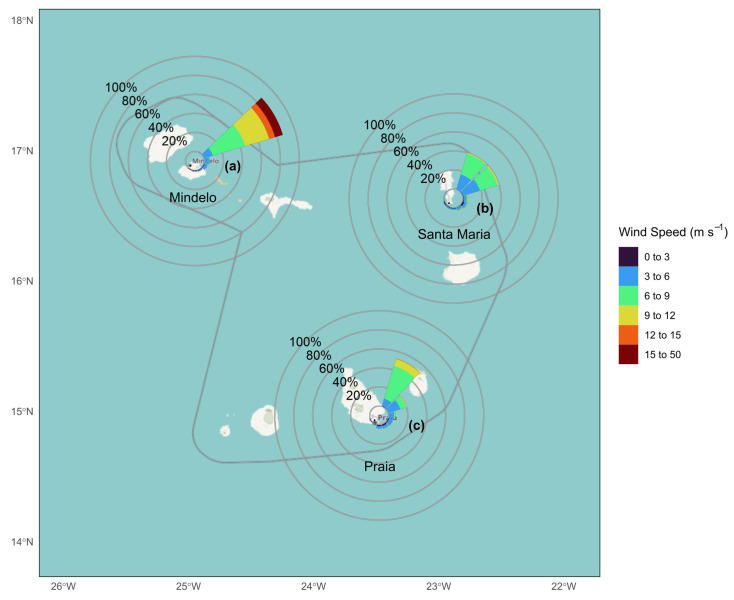
Wind roses for the cities of (**a**) Mindelo at São Vicente island; (**b**) Santa Maria at Sal island; and (**c**) Praia at Santiago island, showing the distribution of wind direction and wind speed in 2023. The wind rose represents the frequency, in percentage, of each wind direction, providing insight into the predominant wind patterns in each island.

**Figure 6 sensors-24-07656-f006:**
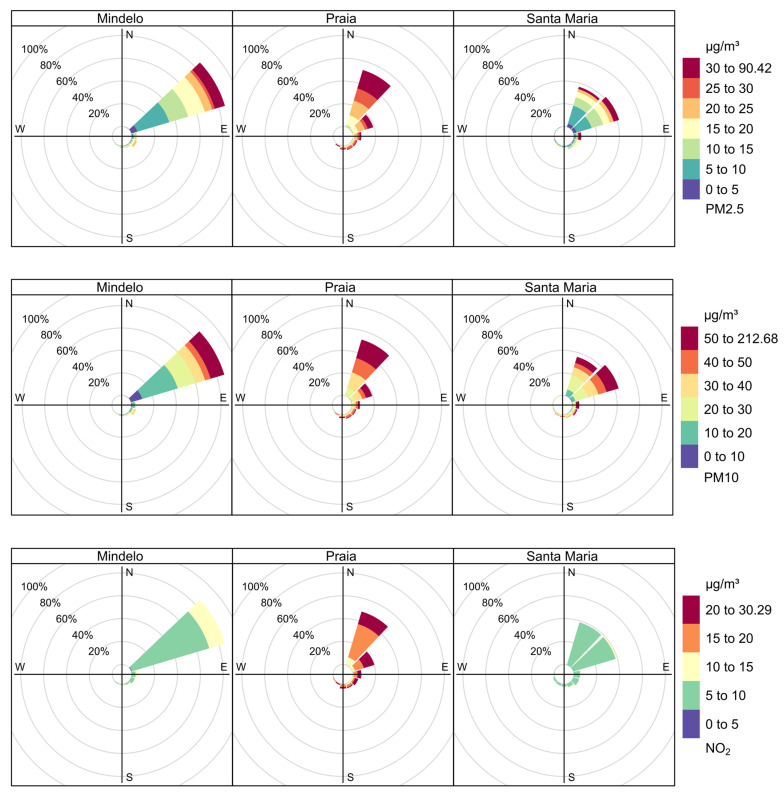
Pollution roses of PM2.5, PM10, and NO_2_ in Mindelo, Praia, and Santa Maria. Each pollution rose shows the frequency of pollutant concentrations by wind direction, with color bands representing different concentration levels.

**Figure 7 sensors-24-07656-f007:**
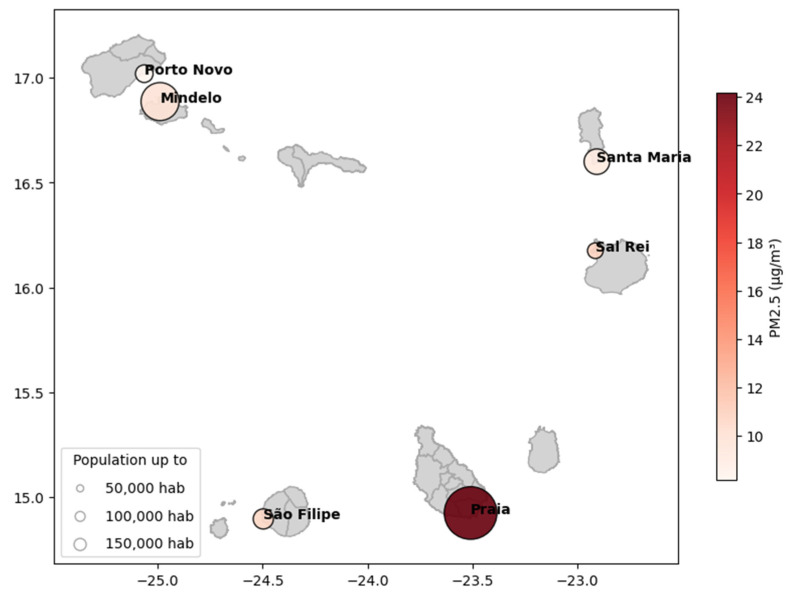
A map of the spatial distribution of the average annual PM2.5 concentrations across the studied cities in Cabo Verde for 2023. The size of each marker represents the population size. The color gradient indicates the PM2.5 concentration.

**Figure 8 sensors-24-07656-f008:**
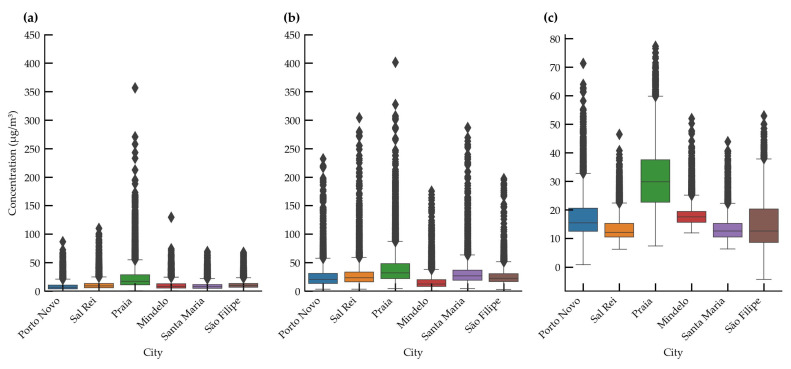
Boxplots of pollutant concentration for different cities in Cabo Verde in 2023: (**a**) PM2.5, (**b**) PM10, and (**c**) NO_2_. The box represents the interquartile range (the middle 50% of the data) with a line at the median. The horizontal lines below and above the box represent the minimum and maximum values within 1.5 times the interquartile range, and the points represent the outliers.

**Figure 9 sensors-24-07656-f009:**
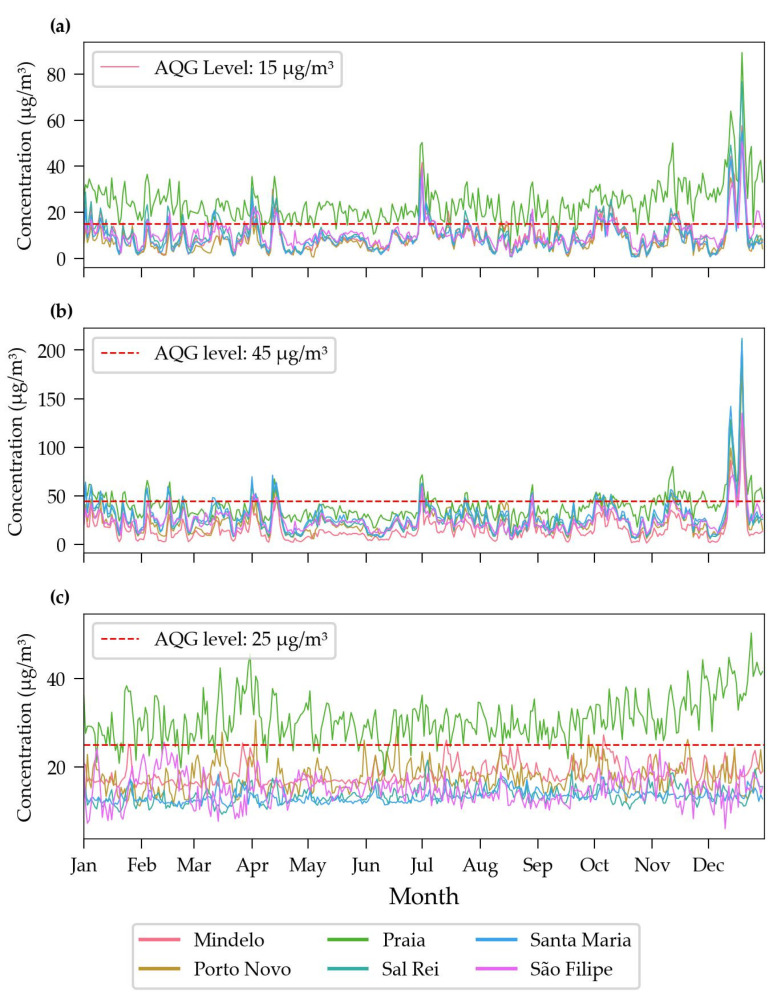
Variation of daily average concentrations of air pollutants compared to WHO global air quality guideline levels (AQG; red dashed line) by location over 2023: (**a**) PM2.5, (**b**) PM10, and (**c**) NO_2_.

**Figure 10 sensors-24-07656-f010:**
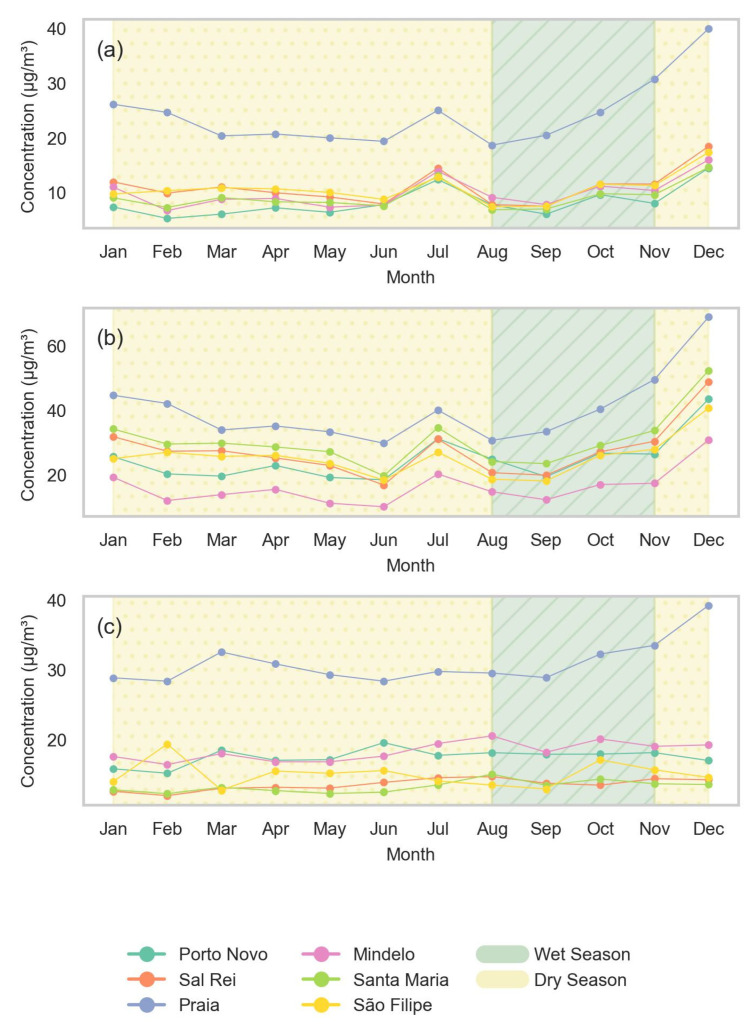
The monthly average concentrations of air pollutants in different locations in Cabo Verde throughout the year 2023: (**a**) PM2.5, (**b**) PM10, and (**c**) NO_2_. The shaded areas represent the dry and wet seasons.

**Figure 11 sensors-24-07656-f011:**
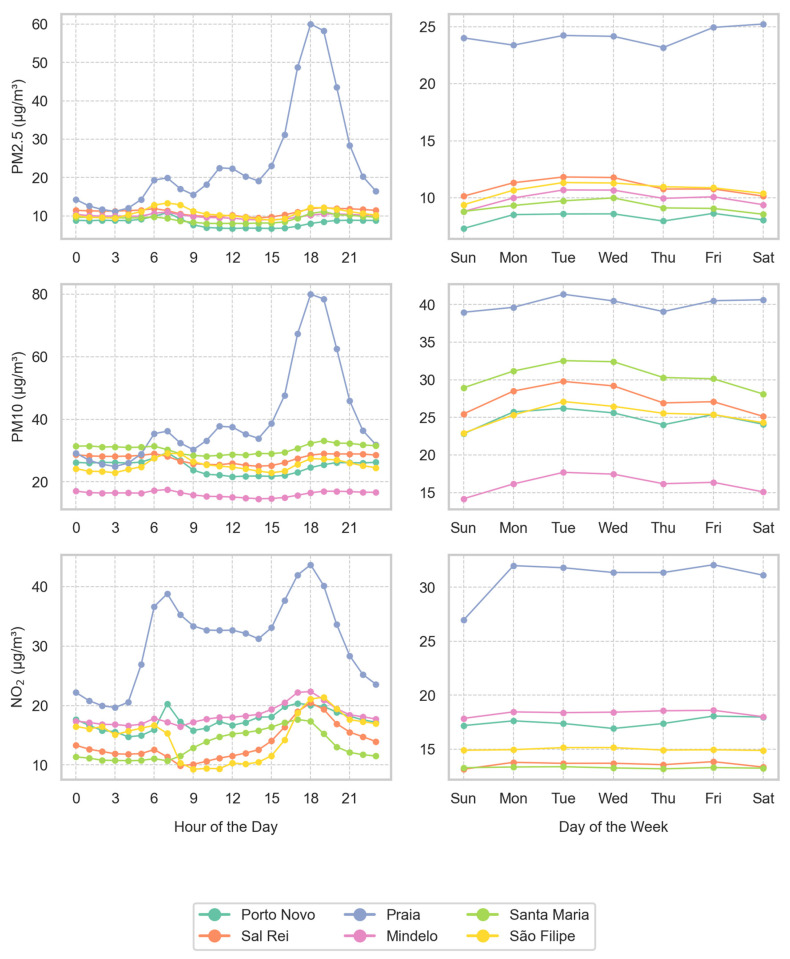
Daily and weekly variations in air pollutant concentrations (PM2.5, PM10, and NO_2_) for all study locations in 2023. Left column: average hourly variations. Right column: average weekly variations.

**Table 1 sensors-24-07656-t001:** Location of the sensors.

Island	City	Location of the Sensors		Sea Level Elevation (m)	Surrounding Characteristics
		Latitude	Longitude		
Santiago	Praia	14.92445	−23.50927	12±	Busy urban area, close to main roads and high traffic zones.
Sal	Santa Maria	16.599833	−22.907778	3±	Coastal area near tourist attractions, moderate traffic.
Santo Antão	Porto Novo	17.019283	−25.064283	7±	Small urban center with less traffic, close to the port area.
São Vicente	Mindelo	16.886325	−24.989092	5±	Urban area, near a busy port and commercial activities.
Boa Vista	Sal Rei	16.175278	−22.915028	6±	Coastal town, low traffic but increasing tourism.
Fogo	São Filipe	14.896028	−24.497056	94±	Small urban city, residential and commercial areas, moderate traffic levels.

**Table 2 sensors-24-07656-t002:** Main characteristics of sensors implemented in network [23].

Parameter	Technology	Range	Performance After Calibration
Particulate Matter [µg/m^3^]	Laser Light Scattering with Remote Calibration	0–1000 µg/m^3^	Accuracy: <100 µg/m^3^: ±10 µg/m^3^; ≥100 µg/m^3^: within ± 10% of measured value. Correlation (R2) with USEPA FEM instrument > 0.8.
		1 µg/m^3^ resolution	
Nitrogen Dioxide NO_2_ [ppb]	Electrochemical Cell with Remote Calibration	0–3000 ppb	Accuracy: <200 ppb: ±30 ppb; ≥200 ppb: ±15% of measured value. Correlation (R2) with USEPA FEM instrument > 0.7.
		1 ppb resolution	

**Table 3 sensors-24-07656-t003:** Average annual concentration (µg/m^3^) for all studied locations.

City	Population	PM2.5	PM10	NO_2_
Mindelo	74,016	9.86	16.08	8.17
Porto Novo	15,914	8.17	24.76	8.27
Praia	142,009	24.16	40.10	18.12
Sal Rei	12,613	10.90	27.37	8.91
Santa Maria	33,347	9.15	30.44	7.24
São Filipe	20,732	10.67	25.24	7.09

**Table 4 sensors-24-07656-t004:** Number of daily exceedances of WHO AQG level for PM2.5, PM10, and NO_2_ for all studied locations over 2023.

City	PM2.5	PM10	NO_2_
Mindelo	61	13	5
Porto Novo	30	26	9
Praia	336	94	331
Sal Rei	67	36	1
Santa Maria	48	48	0
São Filipe	53	18	2

## Data Availability

The data that support the findings of this study are available from the corresponding author, Myriam Lopes, on reasonable request.

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
