# Peer review of "Low-Cost Sensor Network for Air Quality Assessment in Cabo Verde Islands"

_sensors, 2024, doi:10.3390/s24237656_

Round 1
Reviewer 1 Report
Comments and Suggestions for Authors
See the attached 4-page document.

Reviewer 2 Report
Comments and Suggestions for Authors
Paper has limited novelty and relay on mostly descriptive statistics.
It looks like many part of text were generated with AI as there are plenty of typical generative language models words and sentences constructions. This is not allowed by the journal.
Paper needs extensive major revision - especially the scientific novelty should be clearly introduced and summarized.
Specific comments:
Line 52 - please avoid combined citations - if all 10 papers are important please shortly mention the main outcome of each.
Line 59 - really? I think that maintaining costs in developing countries may be lower as workforce is cheaper… please re-write
Line 68 - as this is important to show the successful applications and real live example - please mention few application worldwide of dense LCS measurements - like Cogiel, Weglinska et al. about use of the LCS and ML for spatio-temporal analysis in Krakow and Xing, Sun about meteorological influences with dense network in Beijing, and similar in USA and LA
Figure 1 please add X and Y axis titles
You wrote “The sensor deployment was strategically planned to ensure comprehensive coverage” please describe it more clearly - what was the strategy. Please clarify what means comprehensive coverage while you are using only 6 sensors.
Line 129 131- what is the point of introducing full name of units that are commonly known?
Figure 2 - I assume that the picture is done in one of the studied area mentioned in table so please add it to the figure description
You wrote “To ensure a comprehensive and representative assessment of air quality in the Cabo Verde Islands, the data collection methodology was meticulously planned to cover the entire year from January to December 2023.” - again I’m curious in detail about comprehensiveness and meticulousness in detailed description
Line 151 - please describe how was done the local and regional analysis
Line 163 - again please describe in details the comprehensiveness of the methodology where the meteo parameters where measured in different location than LCS stations
Line 173 - please add citation to Python and R
Line 185-189 it would be good to see the terrain etc to support your statement.
Line 228 - please clarify what authors sees as distinct here
Line 247-252 based on one sensor in the city it is not possible to support that interpretation as fact - rather something that you would like to check next
Line 262 or low cost sensor measurement error….
Also figure 9 - did u consider systematic shift in readings?
Line 295-303 should be moved to introduction
Conclusions without separate discussion are to short and vague
Round 2
Reviewer 1 Report
Comments and Suggestions for Authors
All good. Congrats.
Author Response
Thank you for your comments and suggestions. It has really helped us to improve our work.
Reviewer 2 Report
Comments and Suggestions for Authors
In general paper was improved.
Regarding translation, it is good to ask a native speaker to read the paper beforehand.
1. Maintenance cost response - my comment was regarding the costs and you pointed to developing countries. It doesn't matter if the country is developed or not - the cost of a reference station is higher and LCS is lower.
You wrote in the replay: "Sensors have been placed to cover all main inhabited islands, avoiding local air pollution hotspots, and allowed for data collection with minimal missing data (by being easily accessible for hardware maintenance)" while in the text you wrote:
Line 125: The sensor deployment was strategically planned to ensure coverage of all main urban areas, with sensors installed on the busiest streets of each city.
Please clarify it.
Regarding the part of my comment that you didn't understand - I was referring to that part: "The more pronounced fluctuations in Praia can be attributed to specific local 186 factors, such as urban density and intense human activities, while Sal and São Vicente, 187 with more moderate variations, suggest environments less influenced by urban factors" and my question was if you consider also the topography (terrain) of the islands as a possibly important factor or not.
Where is the nearest reference station? Did you perform a comparison of yours measurements with it? It would be good to add such graph.
Comments on the Quality of English LanguageBesides automatic tool it is good to give text for native speaker before publication.
